# Synthesis of Bioactive Silver Nanoparticles Using New Bacterial Strains from an Antarctic Consortium

**DOI:** 10.3390/md20090558

**Published:** 2022-08-31

**Authors:** Maria Sindhura John, Joseph Amruthraj Nagoth, Kesava Priyan Ramasamy, Alessio Mancini, Gabriele Giuli, Cristina Miceli, Sandra Pucciarelli

**Affiliations:** 1School of Biosciences and Veterinary Medicine, University of Camerino, 62032 Camerino, Italy; 2Department of Dermatology, University of Texas Southwestern Medical Center, Dallas, TX 75390, USA; 3School of Sciences and Technology, University of Camerino, 62032 Camerino, Italy

**Keywords:** nanomaterials, green synthesis, capped nanoparticles, antimicrobial activity, nosocomial pathogens

## Abstract

In this study, we report on the synthesis of silver nanoparticles (AgNPs) achieved by using three bacterial strains *Rhodococcus*, *Brevundimonas* and *Bacillus* as reducing and capping agents, newly isolated from a consortium associated with the Antarctic marine ciliate *Euplotes focardii*. After incubation of these bacteria with a 1 mM solution of AgNO_3_ at 22 °C, AgNPs were synthesized within 24 h. Unlike *Rhodococcus* and *Bacillus*, the reduction of Ag^+^ from AgNO_3_ into Ag^0^ has never been reported for a *Brevundimonas* strain. The maximum absorbances of these AgNPs in the UV-Vis spectra were in the range of 404 nm and 406 nm. EDAX spectra showed strong signals from the Ag atom and medium signals from C, N and O due to capping protein emissions. TEM analysis showed that the NPs were spherical and rod-shaped, with sizes in the range of 20 to 50 nm, and they were clustered, even though not in contact with one another. Besides aggregation, all the AgNPs showed significant antimicrobial activity. This biosynthesis may play a dual role: detoxification of AgNO_3_ and pathogen protection against both the bacterium and ciliate. Biosynthetic AgNPs also represent a promising alternative to conventional antibiotics against common nosocomial pathogens.

## 1. Introduction

The Antarctic marine habitat houses several microbes that must face harsh environmental conditions and stresses, including the presence of heavy metals [1]. The study of such microorganisms contributes to the understanding of environmental adaptation and stress response mechanisms. Furthermore, it makes it possible to discover new secondary metabolites that can be exploited in medicine, chemical industries and biotechnology [2].

Nanotechnology is gaining importance as a discipline due to its innovative materials and applications. Nanoparticles (NPs) are becoming the leading topic in nanotechnology. NPs are particles between 1 and 100 nm in diameter. The small dimensions and high surface-area-to-volume ratio confer them with unique chemical and physical properties useful in a variety of applications: metal NPs are widely used as industrial catalysts and in chemical sensing devices, medical applications, cosmetics, and microelectronics [3]. Silver NPs (AgNPs) are the most widely used and have been incorporated in medical devices and as wound dressings, implants and additives in food processing and textile production with self-preservative properties [4]. Furthermore, AgNPs possess general antibacterial and anti-biofilm activities [5]. Due to the increased microbial resistance to multiple antibiotics [6], AgNPs may represent novel and effective antimicrobial agents [7].

Nowadays, nanomaterial production must be performed using cost-effective and eco-friendly processes, without toxic chemicals in the synthesis and purification protocols. Green synthesis of NPs represents a low-cost and environmentally friendly method with advantages over conventional methods that involve toxic solvents. The most important issue for green NP synthesis is selection of media containing nontoxic reducing agents. There are different green methods for nanoparticle synthesis, but the most commonly used employs bacteria because bacteria are usually easy to grow [8,9]. Bacterial AgNP synthesis is considered a defense mechanism against the very reactive silver ions [10]. Bacterial NPs have remarkable potential since they can be easily coated with a lipid/protein capping, which confers physiological solubility and stability [11]. Capped AgNPs are known to exhibit higher antibacterial activity than uncapped AgNPs [12]. Silver-coated antiseptics show broad-spectrum activity and a far lower chance than standard antibiotics of inducing microbial resistance [12].

In this paper, we report on the intracellular synthesis of AgNPs using three bacterial strains, *Brevundimonas*, *Bacillus*, and *Rhodococcus*, newly isolated from a consortium associated with the psychrophilic Antarctic marine ciliate *Euplotes focardii*, which probably represents a new species; for this reason, each strain was named with the suffix ef1 (ef stands for *Euplotes focardii*). *E. focardii* is a free-swimming ciliate endemic to the oligotrophic coastal sediments of Terra Nova Bay in Antarctica [13,14]. It is strictly psychrophile, with peculiar microtubules dynamics [15,16] and biochemical enzyme properties [17,18,19]. The bacterial consortium associated with this organism has been characterized [20,21,22,23,24] and its ability in the synthesis of metal NPs previously reported [25,26,27]. Transformation of metals into NPs may represent a detoxification strategy and contribute to protection from pathogens [10]. While *Bacillus* [28,29,30] and *Rhodococcus* [31,32] are known as AgNP producers, this paper is the first to describe a *Brevundimonas* strain with this ability. We also show that the AgNPs produced by the consortium-associated bacteria possess antimicrobial activity and can be used against common pathogenic microorganisms.

## 2. Results and Discussion

### 2.1. Biosynthesis and UV–Visible Spectroscopy of Ag NPs

We primarily investigated AgNP biosynthesis by monitoring the change in color of 1 mM AgNO_3_ solutions containing the corresponding bacterial strain. A color change from white to brown occurred within 24 h (Appendix A). No color change was observed in the control cultures containing 1 mM AgNO_3_ and the heat-killed bacterial biomass (Appendix A). Visible culture changes from colorless to brown indicate the reduction of silver nitrate into silver NPs due to excitation of the surface plasmon resonance of the silver NPs [26]. *Bacillus* and *Rhodococcus* strains were previously reported to synthesize AgNPs from AgNO_3_: Saravanan et al. [28] reported a color change from pale yellow to brown in *Bacillus megaterium* cultures due to the reduction of aqueous silver ions to AgNPs. Kulkarni et al. [31] reported AgNP production when *Rhodococcus kroppenstedtii* cell extracts were reacted with AgNO_3_. In contrast, this is the first report of a *Brevundimonas* strain able to reduce AgNO_3_.

AgNP synthesis from AgNO_3_ through reduction of Ag^+^ into Ag^o^ was also monitored using UV–Vis spectroscopy (Appendix A). Sharp peaks that could be attributed to the formation of AgNPs were recorded, with maximum absorption at 406 nm for *Rhodococcus*, 405 nm for *Brevundimonas* and 404 nm for *Bacillus* (Appendix A), a lower range than that of around 420–440 nm previously reported for other *Bacillus* and *Rhodococcus* strains [29,30,31,32,33]. The single peaks obtained indicated that the quality of the synthesized silver NPs was good.

### 2.2. Morphology and Chemical Composition of Bacterial AgNPs

We applied energy-dispersive X-ray (EDAX) analysis to determine the chemical composition of the bacterial AgNPs (Figure 1). In all cases, we observed intense signals at 3 keV due to the surface plasmon resonance of silver, which confirmed the presence of AgNPs [34]. We also detected C and O signals in the normal mode (Figure 1 and accompanying tables). Further peaks for Cl and Na for *Rhodococcus* (A); Cl, Na and N for *Brevundimonas* (B); and Cl and N for *Bacillus* (C) were also observed, which were possibly due to emissions from proteins or enzymes present in the culture supernatant or due to the emissions of the capping proteins, accounting for the reduction of Ag+ ions and the stabilization of AgNPs. These results indicate that the chemical compositions of AgNps from *Rhodococcus*, *Brevundimonas*, and *Bacillus* were different.

We confirmed the interactions between the AgNPs and the capping proteins using FTIR measurements (Appendix A). This technique is a powerful tool used to identify the chemical bonds in a molecule by producing an IR spectrum, which is like a molecular fingerprint. The FTIR spectra were obtained in the range between 600 and 4000 cm^–1^. *Brevundimonas* and *Rhodococcus* samples showed similar IR spectra with comparable vibration peaks, whereas that for *Bacillus* appeared to lack intense absorption bands in the region around 3300–3200 cm^–1^. A detailed description of each FTIR measurement and its assignments is provided in the Appendix A. In general, the distinct bands analyzed indicated the presence of –OH, –NH and –CH2 scissor vibrations for aliphatic compounds and C=C bonds inside the biomolecules, together with the carbonyl group (C=O) of the amide functional group. Therefore, the presence of carbonyl and NH groups confirmed the presence of the capping proteins important for AgNP stabilization [35].

AgNP zeta potential measurements showed a negative charge: −32.8 mV, −29.6 mV and −28.1 mV for *Rhodococcus*, *Brevundimonas* and *Bacillus*, respectively (Appendix A). Nanoparticles with severe negative/positive surface charges are stable. A flat-out zeta estimation of ±30 mV is an overall sign that colloidal solutions are highly stable [36]. These findings suggest that the AgNPs obtained from *Rhodococcus*, *Brevundimonas* and *Bacillus* are highly stable.

The presence of metallic AgNPs was confirmed by XRD analysis (Appendix A). More details on the zeta potential and XRD analysis are reported in the Appendix A.

To measure the AgNP size distribution, we used a laser diffraction method with a multiple scattering technique. For this analysis, the AgNP powder was dispersed in water using ultrasonication. The average sizes of the AgNPs synthesized from *Rhodococcus*, *Brevundimonas*, and *Bacillus* were 40.8 nm, 46.7 nm and 40.6 nm, respectively (Figure 2). The presence of a single peak indicated the good quality of the synthesized Ag NPs.

To further determine the AgNP size and surface morphology, we performed SEM (Appendix A) and TEM investigations (Figure 3). SEM results are reported in more detail in the Appendix A. For all bacteria, TEM analysis revealed that the NPs had spherical shapes. All AgNPs appeared to be clustered but not directly in contact with one another, probably due to the presence of capping peptides at the surface. The average sizes of the AgNPs agreed with the DLS results: 41 nm, 46 nm and 42 nm for *Rhodococcus*, *Bacillus*, and *Brevundimonas*, respectively (Appendix A). The average sizes of NPs from other psychrophilic bacteria, such as *Pseudomonas antarctica*, *Pseudomonas proteolytica*, *Pseudomonas meridiana*, and *Bacillus cecembensis* [37], have been reported to be smaller, i.e., between 6.1 and 12.2 nm.

### 2.3. Antibacterial and Antifungal Activities of Bacterial Ag NPs

A Kirby–Bauer disk diffusion test was performed to compare the AgNPs from *Marinomonas* ef1 with those from *Pseudomonas* ef1 (Figure 4), which were also isolated from the same bacterial consortium and produced non-aggregated NPs [25,26]. A clear inhibition zone around the disks of the *Bacillus*, *Rhodococcus* and *Brevundimonas* AgNPs suggested that, even if aggregated, they possessed antibacterial activity that could inhibit the growth of bacterial and fungal pathogens. The results for the disk diffusion test and the MIC and MBC values for the AgNPs are summarized in Table 1 and Table 2.

### 2.4. MIC and MBC Assay

MIC values for *Brevundimonas* AgNPs are reported in Appendix A and Table 2. Among the Gram-negative bacteria, *Proteus mirabilis* and *Serratia marcescens* showed the lowest MIC value of 6.25 μg/mL, whereas *Escherichia coli* showed the highest MIC value of 25 μg/mL. *Klebsiella*
*pneumoniae*, *Pseudomonas aeruginosa*, *Citrobacter koseri*, and *Acinetobacter baumanii* showed the MIC value of 12.5 μg/mL, whereas the Gram-positive bacteria *Staphylococcus aureus* showed an MIC value of 25 μg/mL. Among fungi, the MIC value for *Candida parapsilosis* and *Candida albicans* was 12.5 μg/mL.

MBC values for *Brevundimonas* are reported in Table 2. Among the Gram-negative bacteria, *Proteus mirabilis* and *Serratia marcescens* showed the lowest MBC value of 12.5 μg/mL. *Escherichia coli*, *Klebsiella pneumoniae*, *Pseudomonas aeruginosa*, *Citrobacter koseri*, and *Acinetobacter baumanii* showed the highest MBC value of 25 μg/mL. Among the Gram-positive bacteria, *Staphylococcus aureus* showed an MBC value of 25 μg/mL, as did *Candida*
*albicans* and *Candida parapsilosis*.

MIC values for *Rhodococcus* AgNPs are shown in Appendix A and Table 2. Among the Gram-negative bacteria, *Serratia*
*marcescens* showed the lowest MIC value of 3.12 μg/mL, whereas *Proteus mirabilis* showed an MIC value of 6.25 μg/mL. Other bacteria, such as *Escherichia coli*, *Klebsiella pneumoniae*, *Pseudomonas aeruginosa*, *Citrobacter koseri*, and *Acinetobacter baumanii*, showed an MIC value of 12.5 μg/mL. Among the Gram-positive bacteria, *Staphylococcus aureus* showed an MIC value of 25 μg/mL, as did *Candida parapsilosis* and *Candida*
*albicans*. MBC values for *Rhodococcus* are reported in Table 2. The lowest MBC value among the Gram-negative bacteria of 12.5 μg/mL was shown by *Proteus mirabilis*, *Citrobacter koseri*, *Acinetobacter baumanii*, and *Serratia*
*marcescens*. *Escherichia coli*, *Klebsiella pneumoniae*, and *Pseudomonas aeruginosa* showed an MBC value of 25 μg/mL. Among the Gram-positive bacteria, *Staphylococcus aureus* showed an MBC value of 25 μg/mL, as did *Candida*
*albicans* and *Candida parapsilosis.*

MIC values for *Bacillus* AgNPs are shown in Appendix A and Table 2. Among the Gram-negative bacteria, *Serratia*
*marcescens* showed the lowest MIC value of 3.12 μg/mL, whereas that of *Proteus mirabilis* was 6.25 μg/mL. *Escherichia coli*, *Klebsiella pneumoniae*, *Pseudomonas aeruginosa*, *Citrobacter koseri*, and *Acinetobacter baumanii* showed an MIC value of 12.5 μg/mL. Among the Gram-positive bacteria, *Staphylococcus aureus* showed an MIC value of 12.5 μg/mL. Among the fungi, the lowest MIC value was noted for *Candida parapsilosis* at 6.25 μg/mL, whereas that for *Candida albicans* was 12.5 μg/mL.

MBC values for Bacillus are reported in Table 2. The lowest MBC value of 12.5 μg/mL was recorded for *Escherichia coli*, Pseudomonas aeruginosa, Proteus mirabilis, Citrobacter koseri, Acinetobacter baumanii, and Serratia marcescens, while Klebsiella pneumoniae showed an MBC value of 25 μg/mL (Gram-negative bacteria). Staphylococcus aureus (Gram-positive bacteria) showed an MBC value of 12.5 μg/mL, as did Candida parapsilosis, whereas that for Candida albicans was 25 μg/mL.

The antimicrobial mechanism of AgNPs is still under debate. A possible process is that AgNPs attach to the pathogen cell wall and/or membrane, modifying their integrity and permeability and disturbing cell respiration [38]. In this case, the antimicrobial activity would depend on the size of the AgNPs: smaller AgNPs function more efficiently than larger ones, probably because they have a higher available surface area for interactions with the cell wall and membrane. Other proposed mechanisms either rely on AgNPs entering the the bacteria cells [39] or on the release of Ag+ from the nanoparticles, since silver ions also have a relevant role in the bactericidal effect [40]. Biosynthesized AgNPs may play a new, key role in pharmacotherapeutics, as they represent one promising approach to overcoming bacterial resistance.

## 3. Materials and Methods

### 3.1. Culture and Chemicals

The bacterial strains *Rhodococcus*, *Brevundimonas* and *Bacillus* were isolated from a consortium associated with the Antarctic ciliate *E. focardii*. All strains have been deposited at the Istituto Zooprofilattico Sperimentale della Lombardia e dell’Emilia Romagna “Bruno Ubertini”—IZSLER in accordance with the Budapest treaty under access numbers *Rhodococcus* sp. ef1 DPS RE RSCIC 4, *Brevundimonas* sp ef1 DPS RE RSCIC 23 and *Bacilus* sp ef1 DPS RE RSCIC 24. Bacterial strains were maintained on Luria–Bertani agar (LB agar) Petri dishes (Tryptone 10 g/L, yeast extract 5 g/L, NaCl 5 g/L) at 22 °C (optimum growth temperature). Analytical grade AgNO_3_ was purchased from Sigma Aldrich (Milan, Italy).

### 3.2. AgNP Biosynthesis and Purification

Single colonies of each individual bacterial strain from overnight LB agar Petri dishes served as the inocula for 100 mL LB broth cultures in 500 mL Erlenmeyer flasks. The cultures were incubated at 22 °C in a shaker (200 rpm) for 24 h. AgNP biosynthesis was carried out by resuspending 200 mg of each bacterial strain in 100 mL of deionized water. A total of 1 mM of AgNO_3_ was added to each culture and the reaction mixture was incubated for 48 h in a rotator shaker at 150 rpm and 22 °C. As a control, the heat-killed biomass of each strain was maintained with 1 mM AgNO_3_ water solution.

To collect the bacterial biomass after 48 h of incubation, the cultures were centrifuged at 4000 rpm for 30 min. The resulting pellet was then suspended in ddH2O and ultra-sonicated at a pulse rate of 6V at intervals of 30 s for 10 cycles (Cheimika, Milan, Italy). The suspension was centrifuged again at 4000 rpm for 30 min (Beckman J2–21, Fullerton, California) to remove cell debris and the supernatant loaded onto a Sephadex G-50 resin equilibrated in 10 mM Tris buffer (pH 7.0) to remove additional contaminating debris and proteins. AgNPs were recovered from the buffered solution by adding isopropanol (three times the initial volume), which is known to dissolve a wide range of non-polar compounds. The mixture was kept in an orbital shaker overnight. After AgNP precipitation and removal of the supernatant, the pellet was subjected to evaporation to obtain a purified powdered highly enriched in AgNPs.

### 3.3. AgNP Characterization: UV–Vis Absorption Spectra and Zeta Potential Measurements

AgNP UV–Vis absorption spectra were recorded in the wavelength range from 200 to 800 nm at room temperature (25 °C) using a Shimadzu UV 1800 spectrophotometer. The size distribution of the particles was determined by measuring the dynamic fluctuations of the light scattering intensity caused by the Brownian motion of the particles (zeta potential) with a Zetasizer Nano ZS (Malvern Instruments Ltd., Malvern, UK). All measurements were carried out in triplicate, with a temperature equilibration time of 2 min at 25 °C.

#### 3.3.1. Scanning Electron Microscopy (SEM) and Transmission Electron Microscopy (TEM)

Scanning Electron Microscope (SEM) analysis was performed using a ZEISS Sigma 300. Purified AgNPs were sonicated for 15 min to obtain a uniform distribution. A drop of the solution was loaded onto carbon-coated copper grids and allowed to evaporate under infrared light for 30 min. AgNP morphologies were observed on a JEOL TEM system operating at 200 kV (JEM-2100, Hitachi Limited, Tokyo, Japan). The particle size was estimated from TEM micrographs using the software Nano Measurer 1.2.5.

#### 3.3.2. Dynamic Light Scattering and Zeta Potential Measurements

A Zetasizer Nano ZS (Malvern Instruments Ltd., Malvern, UK) was used to determine the size distribution of the particles by measuring the dynamic fluctuations of the light scattering intensity caused by the Brownian motion of the particles. All measurements were carried out in triplicate with a temperature equilibration time of 2 min at 25 °C. Additionally, the NP surface charge was measured using the zeta potential [41].

#### 3.3.3. Energy-Dispersive X-ray Analysis (EDAX), Fourier-Transform Infrared Spectroscopy (FTIR) Analysis and Powder X-ray Diffraction (XRD)

The AgNPs’ crystal structure was analyzed using powder X-ray diffraction (XRD) measurements from a Rigaku-D/MAX-PC 2500 X-ray diffractometer with a Cu Kα (λ = 1.5405 Å) in the 2θ range from 20 to 80° at a scan rate of 0.03° S^−1^. FTIR spectra were recorded with a Shimadzu IR Affinity-1 to identify the possible interactions between AgNPs and biomolecules. Analysis was carried out in the range from 400 to 4000 cm^−1^ at a resolution of 4 cm^−1^.

### 3.4. Kirby–Bauer Disk Diffusion Susceptibility Test

Analysis of the antibacterial activity of AgNPs was carried out using a Kirby–Bauer disk diffusion susceptibility test [42,43]. The pathogen strains used are reported in Table 2. The antibacterial activity was tested against *Staphylococcus aureus*, *Escherichia coli*, *Klebsiella pneumoniae*, *Pseudomonas aeruginosa*, *Proteus mirabilis*, *Citrobacter koseri*, *Acinetobacter baumanii*, *Serratia marcescens*, *Candida albicans*, and *Candida parapsilosis*. All the strains were cultured in Mueller Hinton broth (MHB) (Merck, Germany) at 37 °C for 24 h with 200 rpm agitation and then spread on Mueller–Hinton agar (MH agar) using a sterile cotton swab. Paper disks were loaded with 25 μL (25 μg) of AgNPs and AgNO_3_ solution, placed on the agar plate and incubated at 37 °C for 24 h. The zone of inhibition was observed after 24 h of incubation.

### 3.5. Minimum Inhibitory Concentration (MIC) and Minimum Bactericidal Concentration (MBC) Evaluation

The AgNP MIC and MBC values were estimated using the method described in the CLSI 2012 guideline [44].

The MIC test was performed in 96-well round-bottom microtiter plates using standard broth microdilution methods. The MBC test was performed with MH agar plates. Bacteria concentrations were adjusted to 0.5 McFarland units. For the MIC test, AgNP stock solutions of 200 μg/mL were prepared by dispersing AgNPs in sterilized deionized water using ultrasonics: 100 μL of the stock solution was serially diluted twofold in the first row containing 100 μL of MHB, and then 100 μL was discarded from the last well so that the first well in the row of the microtiter plate contained the highest AgNP concentration and the last well in the row contained the lowest concentration. AgNPs dilutions were prepared in all the other rows following the same protocol. The negative control contained only medium (K−) and the positive control contained medium and bacterial inocula (K+). The microtiter plate was then incubated at 37 °C for 24 h. The MIC values were defined as the lowest AgNP concentrations that inhibited bacterial growth.

To check the MBC values, the suspensions from each well of the microtiter plates were transferred into MH agar plates and incubated at 37 °C for 24 h. The MBC values were defined as the lowest concentration of the antibacterial agents that completely killed the bacteria.

## 4. Conclusions

We described an easy and efficient biological method to synthesize AgNPs using the novel strains named *Rhodococcus* ef1, *Brevundimonas* ef1, and *Bacillus* ef1, isolated from a bacterial consortium found in association with the Antarctic psychrophilic ciliate *E. focardii*. The present synthesis method is an efficient and eco-friendly alternative to chemical protocols and can be explored for large-scale production. *Rhodococcus*, *Brevundimonas*, and *Bacillus* AgNPs were characterized and found to be stable and uniform and to have high antimicrobial activity comparable with that of *Marinomonas* ef1 and *Pseudomonas* ef1, the other two strains isolated from the Antarctic consortium. The study highlighted an efficient strategy to obtain bionanomaterials that can be used with many drug-resistant pathogens, contributing to addressing the global concern regarding antibiotic-resistant bacteria. The use of biosynthesized AgNPs represents a more efficient and promising substitute for conventional antibiotics than non-nanometric silver. Finally, *Rhodococcus*, *Brevundimonas*, and *Bacillus* strains can also be exploited for bioremediation to remove silver nitrate contamination from the environment.

## Figures and Tables

**Figure 1 marinedrugs-20-00558-f001:**
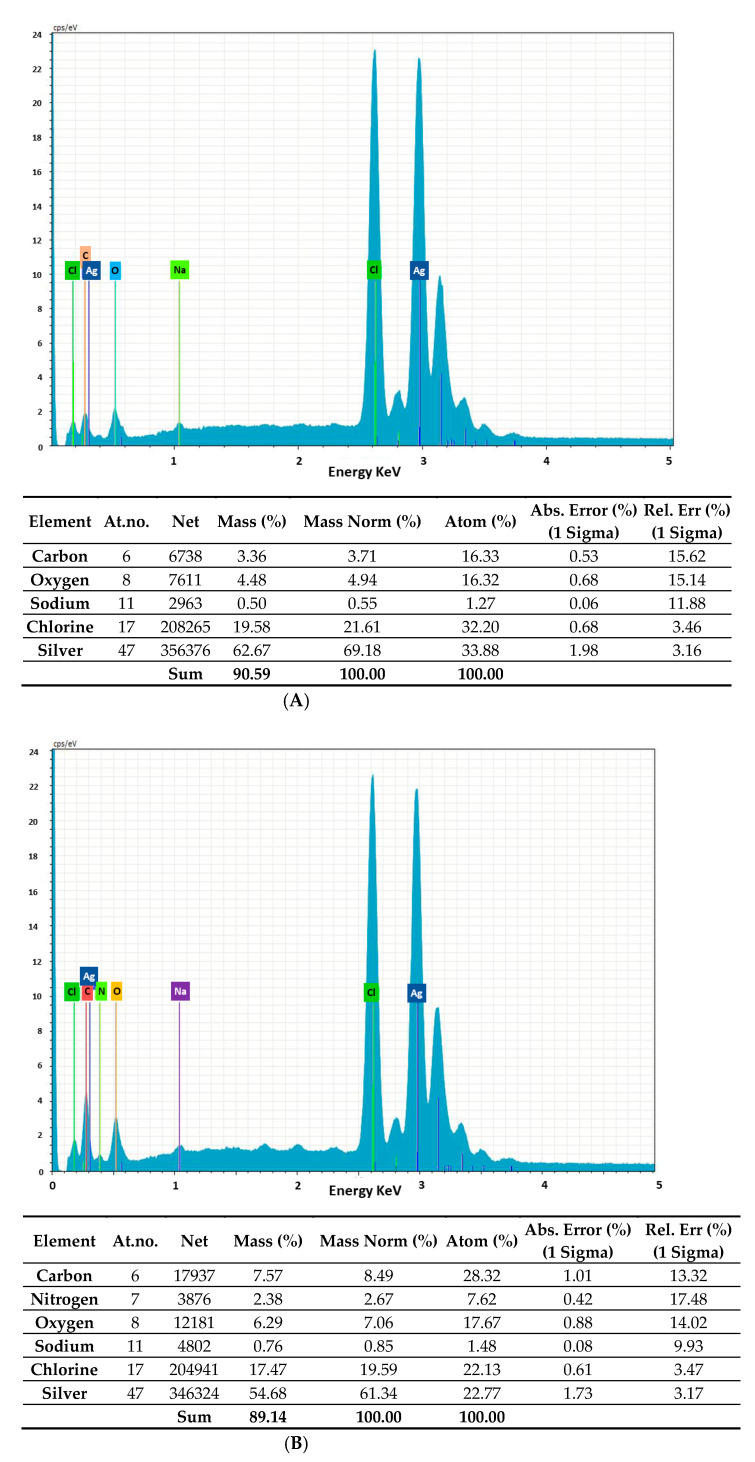
EDAX analysis of AgNPs synthesized from *Rhodococcus* (**A**), *Brevundimonas* (**B**) and *Bacillus* (**C**). Ag, C, N and O indicate the silver (the highest peak, recorded at 3 or 3.2 keV), carbon, nitrogen and oxygen signals (the relative amounts are reported in the tables).

**Figure 2 marinedrugs-20-00558-f002:**
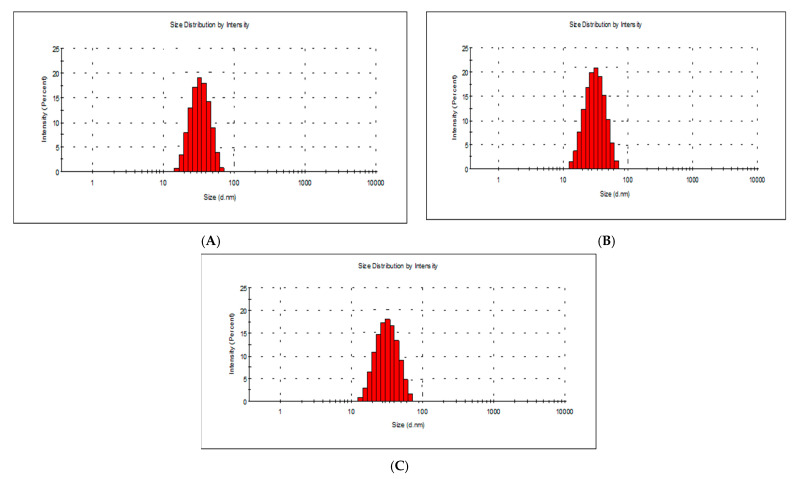
Dynamic light scattering particle size analysis of *Rhodococcus* (**A**), *Brevundimonas* (**B**) and *Bacillus* (**C**) AgNPs. The average particle sizes were 40.8 nm, 46.7 nm and 40.6 nm, respectively.

**Figure 3 marinedrugs-20-00558-f003:**
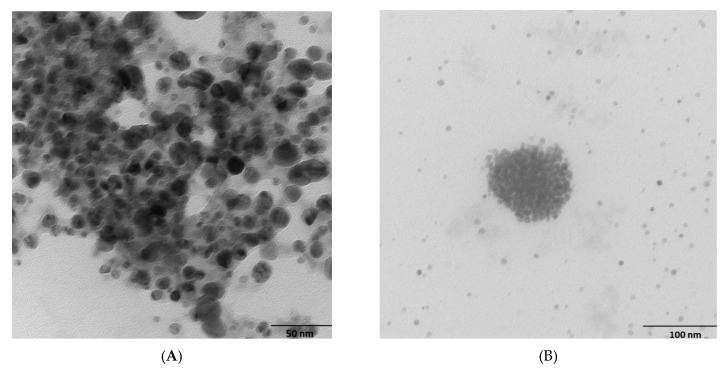
TEM images of biosynthesized AgNPs from *Rhodococcus* (**A**), *Bacillus* (**B**), and *Brevundimonas* (**C**).

**Figure 4 marinedrugs-20-00558-f004:**
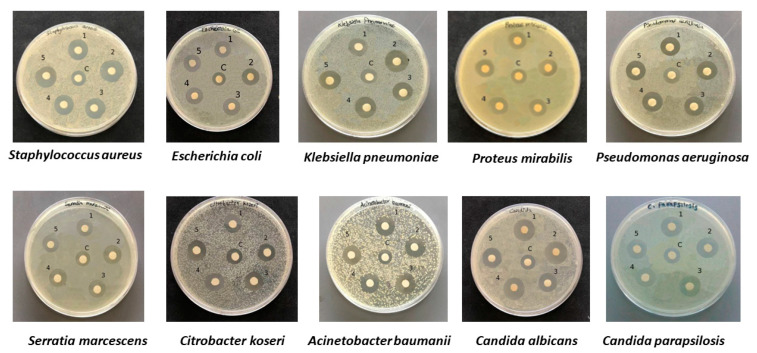
Kirby–Bauer test performed with AgNPs from 1. *Marinomonas* ef1; 2. *Rhodococcus* ef1; 3. *Brevundimonas* ef1; 4. *Pseudomonas* ef1; 5. *Bacillus* ef1. C indicates the control represented by 1 mM AgNO_3_. The test was performed with *Staphylococcus aureus*, *Escherichia coli*, *Klebsiella pneumoniae*, *Proteus mirabilis*, *Pseudomonas aeruginosa*, *Serratia marcescens*, *Citrobacter koseri*, *Acinetobacter baumanii*, *Candida albicans*, and *Candida parapsilosis*.

**Table 1 marinedrugs-20-00558-t001:** Variations in bacterial and fungal pathogen inhibitions zones against bacterial AgNPs or AgNO_3_.

Antibacterial Activity of Bio AgNPs																
Pathogenic Bacteria	*Marinomonas* ef1			*Pseudomonas* ef1			*Rhodococcus* ef1			*Brevundimonas* ef1			*Bacillus* ef1			
	Diameter of Inhibition Zone (mm)															
	T^1^	C^1^	IZS^3^ (T-C)	T^1^	C^1^	IZS^3^ (T-C)	T^1^	C^1^	IZS^3^ (T-C)	T^1^	C^1^	IZS^3^ (T-C)	T^1^	C^1^	C^2^	IZS^3^ (T-C)
**GRAM-POSITIVE BACTERIA**																
*Staphylococcus aureus*	15 ± 0.2	9 ± 0.4	6 ± 0.2	13 ± 0.1	9 ± 0.2	4 ± 0.1	16 ± 0.3	9 ± 0.2	7 ± 0.1	16 ± 0.1	9 ± 0.4	7 ± 0.3	15 ± 0.2	9 ± 0.4	17 ± 0.4	6 ± 0.2
**GRAM-NEGATIVE BACTERIA**																
*Escherichia coli*	17 ± 0.3	10 ± 0.4	7 ± 0.1	16 ± 0.2	10 ± 0.1	6 ± 0.1	16 ± 0.2	10 ± 0.4	6 ± 0.2	17 ± 0.2	10 ± 0.1	7 ± 0.1	16 ± 0.5	10 ± 0.2	9 ± 0.2	6 ± 0.3
*Klebsiella pneumoniae*	16 ± 0.5	10 ± 0.2	6 ± 0.3	16 ± 0.3	10 ± 0.5	6 ± 0.2	15 ± 0.2	10 ± 0.3	5 ± 0.1	17 ± 0.5	10 ± 0.4	7 ± 0.1	16 ± 0.1	10 ± 0.3	10 ± 0.5	6 ± 0.2
*Pseudomonas aeruginosa*	15 ± 0.2	11 ± 0.5	4 ± 0.3	15 ± 0.2	11 ± 0.4	4 ± 0.2	17 ± 0.3	11 ± 0.5	6 ± 0.2	16 ± 0.4	11 ± 0.2	5 ± 0.2	15 ± 0.2	11 ± 0.3	15 ± 0.2	4 ± 0.1
*Proteus mirabilis*	14 ± 0.3	10 ± 0.1	4 ± 0.2	15 ± 0.4	10 ± 0.2	5 ± 0.2	15 ± 0.4	10 ± 0.3	5 ± 0.1	14 ± 0.2	10 ± 0.4	4 ± 0.2	15 ± 0.4	10 ± 0.2	12 ± 0.2	5 ± 0.2
*Citrobacter koseri*	15 ± 0.4	10 ± 0.3	5 ± 0.1	16 ± 0.2	10 ± 0.3	6 ± 0.1	15 ± 0.2	10 ± 0.4	5 ± 0.2	15 ± 0.3	10 ± 0.2	5 ± 0.1	16 ± 0.2	10 ± 0.1	R	6 ± 0.1
*Acinetobacter baumanii*	14 ± 0.5	11 ± 0.2	3 ± 0.3	16 ± 0.2	11 ± 0.1	5 ± 0.1	17 ± 0.2	11 ± 0.6	6 ± 0.4	15 ± 0.5	11 ± 0.2	4 ± 0.3	15 ± 0.4	11 ± 0.2	R	4 ± 0.2
*Serratia marcescens*	14 ± 0.3	10 ± 0.5	4 ± 0.2	14 ± 0.2	10 ± 0.1	4 ± 0.1	15 ± 0.4	10 ± 0.1	5 ± 0.3	15 ± 0.2	10 ± 0.1	5 ± 0.1	14 ± 0.4	10 ± 0.3	18 ± 0.2	4 ± 0.1
**FUNGI**																
*Candida albicans*	14 ± 0.4	11 ± 0.1	3 ± 0.3	17 ± 0.6	11 ± 0.5	6 ± 0.1	16 ± 0.4	11 ± 0.3	5 ± 0.1	16 ± 0.2	11 ± 0.3	5 ± 0.1	18 ± 0.2	11 ± 0.3	12 ± 0.5	7 ± 0.1
*Candida parapsilosis*	12 ± 0.1	11 ± 0.2	1 ± 0.1	16 ± 0.5	11 ± 0.3	5 ± 0.2	19 ± 0.2	11 ± 0.1	8 ± 0.1	17 ± 0.4	11 ± 0.1	6 ± 0.3	16 ± 0.5	11 ± 0.3	14 ± 0.4	5 ± 0.1

C^1^ = control (AgNO_3_); C^2^ = control (ampicillin for bacteria, amphotericin B for *Candida*); T^1^ = test (AgNPs); IZS^3^ = increased zone size, obtained from the difference between the mm halo of the control and the test. Data were measured in mm and represent the mean of three experimental values.

**Table 2 marinedrugs-20-00558-t002:** MIC and MBC values of *Rhodococcus* ef1, *Brevundimonas* ef1 and *Bacillus* ef1 AgNPs against pathogenic bacteria.

Pathogenic Bacteria	*Rhodococcus* ef1		*Brevundimonas* ef1		*Bacillus* ef1	
	MIC µg/mL	MBC μg/mL	MIC µg/mL	MBC μg/mL	MIC µg/mL	MBC μg/mL
**GRAM-POSITIVE BACTERIA**						
*Staphylococcus aureus* **(** **ATCC** **V** **@25923)**	25 ± 0.4	25 ± 0.2	25 ± 0.2	25 ± 0.1	12.5 ± 0.2	12.5 ± 0.3
**GRAM-NEGATIVE BACTERIA**						
*Escherichia coli* **(** **ATCC** **V** **@** **25922)**	12.5 ± 0.2	25 ± 0.2	25 ± 0.4	25 ± 0.1	12.5 ± 0.2	12.5 ± 0.4
*Klebsiella pneumoniae* (**ATCC****V****@13883)**	12.5 ± 0.4	25 ± 0.3	12.5 ± 0.4	25 ± 0.2	12.5 ± 0.2	25 ± 0.3
Pseudomonas *aeruginosa* **(ATCC****V****@27853)**	12.5 ± 0.2	25 ± 0.2	12.5 ± 0.2	25 ± 0.3	12.5 ± 0.4	12.5 ± 0.4
*Proteus mirabilis* **(ATCC** **V** **@35659)**	6.25 ± 0.3	12.5 ± 0.5	6.25 ± 0.3	12.5 ± 0.4	6.25 ± 0.4	12.5 ± 0.2
*Citrobacter koseri* **(ATCC** **V** **@ 25408)**	12.5 ± 0.5	12.5 ± 0.4	12.5 ± 0.3	25 ± 0.2	12.5 ± 0.3	12.5 ± 0.2
*Acinetobacter baumanii* **(ATCC** **V** **@ 19606)**	12.5 ± 0.2	12.5 ± 0.2	12.5 ± 0.4	25 ± 0.3	12.5 ± 0.4	12.5 ± 0.2
*Serratia marcescens* **(ATCC** **V** **@13880)**	3.12 ± 0.3	12.5 ± 0.2	6.25 ± 0.3	12.5 ± 0.2	3.12 ± 0.5	12.5 ± 0.5
**FUNGI**						
*Candida albicans* **(ATCC** **V** **@ 90028)**	12.5 ± 0.5	25 ± 0.2	12.5 ± 0.4	25 ± 0.4	12.5 ± 0.3	25 ± 0.3
*Candida parapsilosis* **(ATCC** **V** **@22019)**	12.5 ± 0.4	25 ± 0.5	12.5 ± 0.4	25 ± 0.5	6.25 ± 0.3	12.5 ± 0.4

## Data Availability

Not applicable.

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
