# Peer review of "Synthesis of Bioactive Silver Nanoparticles Using New Bacterial Strains from an Antarctic Consortium"

_marinedrugs, 2022, doi:10.3390/md20090558_

Round 1

Reviewer 1 Report

Sandra Pucciarelli and co-workers reported a novel green synthesis method for silver nanoparticles using three bacterial strains associated with the Antarctic marine ciliate Euplotes focardii. Characterized the nanoparticles and studied their antimicrobial activity against common pathogenic microorganisms. The work is overall well organized and deserves to be published in Marine Drugs. This work is interesting and adds to the existing knowledge of AgNP synthesis method of natural origin.

I appreciate author's efforts in reporting. However, there are multiple errors/inaccurate statements that prevent it from acceptance in the current version.

Overall observations:

Introduction

The authors introduced the topic, context, importance, and hypothesis in a decent manner. Include more recent citations from 2021/22. Example issue: https://www.mdpi.com/journal/ijms/special_issues/materials_medical_applications

Results and Discussion

Poorly written, many statements are vague and incorrect, especially the characterization section.

The supplementary file seems to be created in hurry and doesn’t match the quality of the main manuscript. It is important to treat those files as an integral part of the manuscript. I request authors to check spellings and superscripts for IR data and figure numbers of figure legends carefully.

Materials and methods:

Authors cited just a single reference in the entire method sections. This is not acceptable quality. However, the authors used well-defined methods from previous literature.

Authors used Kirby–Bauer Disk Diffusion Susceptibility Test. No mention of original citation for this well-established method.

Sperate this section into each method (3.3.1, 3.3.2 etc) and properly mention citations for each and solvents/pH/replicates if any. All machine manufacturers and country names must be uniform and accurate.

“3.3. AgNPs characterization: UV/Vis absorption spectra, Scanning Electron Microscopy

(SEM), Transmission Electron Microscopy (TEM) and Energy Dispersive X-ray Analysis (EDAX), Fourier Transform Infrared Spectroscopy (FTIR) Analysis, powder X-ray diffraction (XRD) and Zeta Potential measurements”

 Detailed comments

Major

1. Did you perform the statistics to the zone of inhibitions? For example, Comparisons between different bacterium types would be interesting and important. It will improve the manuscript’s discussion. Conduct statistics for Table 1 and Table 2.

 2. In nanoparticle characterization SD and AVERAGES must be reported.

 3.  235-237 This is one reason for your aggregation shown in SEM/TEM. Did the authors consider the freeze drying technique? or lyophilization process? If you simply evaporate, do you think the aggregation will be higher? Explain this well in discussion and cite proper references.

Minor:

1.     Figure S1, A, C and E) where are these figures?

2.     91-96 poor representation of S2 figure provide A B C format

3.     105 FTIR image number is wrong, and quality must be improved.

4.     Discuss cl peaks as well in EDAX and tables should not be in image format. Poor resolution

a.     Change to A  b. change to B

this has to be change in entire manuscript use same letters in images and text

5.     111-116 check number for figure and correct it

6.     111-115 report averages and SD

7.     AgNPs zeta potential measures showed negative charge which indicated that the samples are stable at room temperature

Why is that negative charge indicates stability? Discuss this in manuscript.

https://pubmed.ncbi.nlm.nih.gov/27297779/

8.     “The obtained single peak indicated that the quality of the synthesized silver NPs was good”

This is not correct. Justify or remodify it after reading literature.

9.     Mie-scattering theory what is reference? Cite it

10. 119-122 The average size of AgNPs synthesized from Rhodococcus, Brevundimonas, and Bacillus were of 40.8nm, 46.7nm and 40.6 nm respectively. The presence of single peak indicates good quality of the synthesized Ag NPs.

Why is that your TEM distribution is different? Provide discussion

Repot mean and SD

Report volume distribution data and number distribution data from zeta sizer machine.

Report Span values for each particle size either in main text or suppli section refer https://www.materials-talks.com/d90-d50-d10-and-span-for-dls/

121 “The presence of single peak indicates good quality of the synthesized Ag NPs.”

The fact is that single peak not amounts to much if you have a broad distribution range on X axis. For example, think someone provides 1-1000 nm particle distribution with single peak. it is good? Authors may put effort and understand this from literature including Malvern resources. Example https://pubmed.ncbi.nlm.nih.gov/27297779/

11. Figure 3 a must A and b B c C

12. 130-133

The average size of the NPs obtained is 50nm, whereas those from other psychrophilic bacteria as Pseudomonas antarctica, Pseudomonas proteolytica, Pseudomonas meridiana, Bacillus cecembensis [35] were reported to be smaller i.e., between 6.1 and 12.2 nm.

Is this because of intensity measurements from DLS? Your TEM values seems lower do you think TEM is better over DLS for correlation?

13. 151 Citrobacter koseri; Acinetobacter baumanii; I cannot read those letters in figure 4 modify it

14. Table 1 and 2 no means or deviations why is that?

15. 195-198 figure 5 is not useful to the reader. Better you give graph in bar format or curve format. Move this figure to suppli data if you need to.

16. Please change all images “a”  A   switched everywhere.

17. 204-205 In this case, the antimicrobial activity dependents on AgNPs size: smaller AgNPs function more efficiently than larger ones, probably because these have a higher available surface area for interactions with the cell wall and membrane.

This is not entirely true. There are threshold values. Refer relevant sources or justify this sentence.

18. 219 country location

232 what source?

233 what do you mean 3 volumes?

243 who made it? All machines should have country and company name and model numbers

273 provide URL

19. 309-314 Why is this as 5? If it is appendix or extra move there. Those numbers are IT patent?

20. 316-320 several errors in this section. Through overhaul is required. 

Author Response

Below are reported all the responses to the referee.

Overall observations:

Introduction

The authors introduced the topic, context, importance, and hypothesis in a decent manner. Include more recent citations from 2021/22. Example issue: https://www.mdpi.com/journal/ijms/special_issues/materials_medical_applications

Response: In the revised version of the paper we cited more recent bibliography

Results and Discussion

Poorly written, many statements are vague and incorrect, especially the characterization section.

The supplementary file seems to be created in hurry and doesn’t match the quality of the main manuscript. It is important to treat those files as an integral part of the manuscript. I request authors to check spellings and superscripts for IR data and figure numbers of figure legends carefully.

Response: In the revised version of the supplementary materials we checked teh spelling and improved the IR figure

Materials and methods:

Authors cited just a single reference in the entire method sections. This is not acceptable quality. However, the authors used well-defined methods from previous literature.

Response:: In the revised version of the paper we cited more references

Authors used Kirby–Bauer Disk Diffusion Susceptibility Test. No mention of original citation for this well-established method.

Response:: In the revised version of the paper we cited the method

Seperate this section into each method (3.3.1, 3.3.2 etc) and properly mention citations for each and solvents/pH/replicates if any. All machine manufacturers and country names must be uniform and accurate.

 “3.3. AgNPs characterization: UV/Vis absorption spectra, Scanning Electron Microscopy

(SEM), Transmission Electron Microscopy (TEM) and Energy Dispersive X-ray Analysis (EDAX), Fourier Transform Infrared Spectroscopy (FTIR) Analysis, powder X-ray diffraction (XRD) and Zeta Potential measurements”

 Response: In the revised version of the paper we separated this paragraph

Detailed comments

Major

  1. Did you perform the statistics to the zone of inhibitions? For example, Comparisons between different bacterium types would be interesting and important. It will improve the manuscript’s discussion. Conduct statistics for Table 1 and Table 2.

Response: in the revised version of the paper we added statistics and comparisons in table 1 and 2

  1. In nanoparticle characterization SD and AVERAGES must be reported.

Response: we added this in the revised version of the paper

  1. 235-237 This is one reason for your aggregation shown in SEM/TEM. Did the authors consider the freeze drying technique? or lyophilization process? If you simply evaporate, do you think the aggregation will be higher? Explain this well in discussion and cite proper references.

Response: We used isopropanol extraction to increase the purity and stability of AgNPs, following other studies on biosynthesis of AgNPS. We believe that nanoparticles are clustered not aggregated. We reported this in the revised version of the paper.

Minor:

  1. Figure S1, A, C and E) where are these figures?

Response: These are in the first figures of the suppelmentary materials

  1. 91-96 poor representation of S2 figure provide A B C format

Response: In the revised version of the paper we improved the quality of S2 figure

  1. 105 FTIR image number is wrong, and quality must be improved.

Response: In the revised version of the paper we improved the quality of FTIR figure

  1. Discuss cl peaks as well in EDAX and tables should not be in image format. Poor resolution

Response: In the revised version of the paper we improved the quality of the EDAX figure and tables,and we better discuss the results

  1. Change to A  b. change to B. This has to be change in entire manuscript use same letters in images and text

Response: done

  1. 111-116 check number for figure and correct it

Response: done

  1. 111-115 report averages and SD
  2. Response: done

  1. AgNPs zeta potential measures showed negative charge which indicated that the samples are stable at room temperature

Why is that negative charge indicates stability? Discuss this in manuscript.

https://pubmed.ncbi.nlm.nih.gov/27297779/

Response: Nanoparticles with severe negative/positive surface charges are stable. A flat-out zeta possible estimation of ±   30 mV is an overall sign that the colloidal solutions are highly stable. AgNPs have a mean size of nanoparticles of 29 nm and a zeta potential of −   18.5 mV value; the CMGK stabilized AgNPs had a zeta potential of −   18.7 mV (Kondaiah et al., 2018). We better explain this in the revised version of the paper

  1. “The obtained single peak indicated that the quality of the synthesized silver NPs was good”

This is not correct. Justify or remodify it after reading literature.

Response: The single peaks was referred to UV spectra not to DLS and Zeta potential. We corrected this statement in the revised version of the paper

  1. Mie-scattering theory what is reference? Cite it

Response: This statement is not correct, We removed it from our manuscript.

  1. 119-122 The average size of AgNPs synthesized from Rhodococcus, Brevundimonas, and Bacillus were of 40.8nm, 46.7nm and 40.6 nm respectively. The presence of single peak indicates good quality of the synthesized Ag NPs.

Why is that your TEM distribution is different? Provide discussion

.Response: the mean values are not so different: 41nm, 45nm and 42nm for Rhodococcus, Bacillus and Brevundimonas.

Repot mean and SD

Response: done

Report volume distribution data and number distribution data from zeta sizer machine.

Response: done

Report Span values for each particle size either in main text or suppli section refer https://www.materials-talks.com/d90-d50-d10-and-span-for-dls/

Response: We reported all these in the revised version of the paper

121 “The presence of single peak indicates good quality of the synthesized Ag NPs.”

The fact is that single peak not amounts to much if you have a broad distribution range on X axis. For example, think someone provides 1-1000 nm particle distribution with single peak. it is good? Authors may put effort and understand this from literature including Malvern resources. Example https://pubmed.ncbi.nlm.nih.gov/27297779/

Response: The single peaks was referred to UV spectra not to DLS and Zeta potential. We corrected in the revised version of the paper

  1. Figure 3 a must A and b B c C

Response: done

  1. 130-133 The average size of the NPs obtained is 50nm, whereas those from other psychrophilic bacteria as Pseudomonas antarctica, Pseudomonas proteolytica, Pseudomonas meridiana, Bacillus cecembensis [35] were reported to be smaller i.e., between 6.1 and 12.2 nm.

 Is this because of intensity measurements from DLS? Your TEM values seems lower do you think TEM is better over DLS for correlation?

Response: The statement wanted to stress that our psychrophilic bacteria are capable of forming nanoparticles comparable with other Antacric bacteria. The average size of NPs from TEM distribution correlates with DLS. They were 41nm, 45nm and 42nm for Rhodococcus, Bacillus and Brevundimonas.

  1. 151 Citrobacter koseri; Acinetobacter baumanii; I cannot read those letters in figure 4 modify it

Response: in our version of the paper the names are clearly readable.

  1. Table 1 and 2 no means or deviations why is that?

Response: in the revised version of the paper we added these values

  1. 195-198 figure 5 is not useful to the reader. Better you give graph in bar format or curve format. Move this figure to suppli data if you need to.

Answer: We moved to supplementary materials.

  1. Please change all images “a”  A   switched everywhere.

Answer: done

  1. 204-205 In this case, the antimicrobial activity dependents on AgNPs size: smaller AgNPs function more efficiently than larger ones, probably because these have a higher available surface area for interactions with the cell wall and membrane.

This is not entirely true. There are threshold values. Refer relevant sources or justify this sentence.

Rsponse: we reported citation. Qing Y, Cheng L, Li R, Liu G, Zhang Y, Tang X, Wang J, Liu H, Qin Y. Potential antibacterial mechanism of silver nanoparticles and the optimization of orthopedic implants by advanced modification technologies. Int J Nanomedicine. 2018 Jun 5;13:3311-3327. doi: 10.2147/IJN.S165125. PMID: 29892194; PMCID: PMC5993028.

  1. 219 country location

Answer: done

232 what source?

Answer: we reported the strains in Table 2

233 what do you mean 3 volumes?

Response: Three times the initial volume. We reported this in the revised version of the paper

243 who made it? All machines should have country and company name and model numbers

response: done

273 provide URL

response done

  1. 309-314 Why is this as 5? If it is appendix or extra move there. Those numbers are IT patent?

response: in these section we reported the international patents deposit number and the bacterial deposit number

  1. 316-320 several errors in this section. Through overhaul is required.

response: we changed as suggested.

Reviewer 2 Report

The manuscript was well written in term of format and its content. Several aspect can be improved for better manuscript. Herein I attach the file with commented and suggested edit regarding its content and format.

Author Response

we accepted all the comments and corrections

Round 2

Reviewer 1 Report

After the revision, the manuscript is appropriate for publication. The authors addressed all the questions of the reviewer, and the answers are satisfactory. Thus, the manuscript can be published in revised form.